# Stunting at birth and associated factors among newborns delivered at the University of Gondar Comprehensive Specialized Referral Hospital

**Almaz Tefera Gonete**[1]*, **Bogale Kassahun**[1], **Eskedar Getie Mekonnen**[2], **Wubet Worku Takele**[3]

1 Department of Pediatrics and Child Health Nursing, School of Nursing, College of Medicine and Health Sciences, University of Gondar, Gondar, Ethiopia, 2 Department of Reproductive and Child Health, Institute of Public Health, College of Medicine and Health Sciences, University of Gondar, Gondar, Ethiopia, 3 Department of Community Health Nursing, School of Nursing College of Medicine and Health Sciences, University of Gondar, Gondar, Ethiopia

* almazteferag3@gmail.com

## Abstract

### Background

Stunting at birth is a chronic form of undernutrition majorly attributable to poor prenatal nutrition, which could persist in children's later life and impact their physical and cognitive health. Although multiple studies have been conducted in Ethiopia to show the magnitude of stunting and factors, all are concentrated on children aged between 6 to 59 months. Therefore, this study was done to determine the prevalence and associated factors of stunting at birth among newborns delivered at the University of Gondar Comprehensive Specialized Referral Hospital, Northwest, Ethiopia.

### Methods

An institution-based cross-sectional study was conducted from February 26th to April 25th/2020. A systematic random sampling technique was used, to select a total of 422 newborn-mother pairs. The binary logistic regression was employed to identify factors associated with stunting and all independent variables were entered into the multivariable logistic regression model to adjust for confounders. Variables that had significant association were identified based on p-value < 0.05 and the adjusted odds ratio with its respective 95% confidence interval was applied to determine the strength as well as the direction of the association.

### Results

About 30.5% (95% CI: 26.3%, 35.1%) of newborns were stunted at birth. Being male [Adjusted odds ratio (AOR) = 2.9(1.62, 5.21)], newborns conceived in Kiremt(rainy season) [AOR = 2.7(1.49, 4.97)], being low birth weight [AOR = 3.1(1.64, 6.06)] were factors associated with stunting at birth. Likewise, newborns born to short stature mothers [AOR = 2.8

**Data Availability Statement:** All relevant data are within the manuscript and its Supporting information files.

**Funding:** The University of Gondar sponsored the study, however, the funder didn't have a role in the study.

**Competing interests:** The authors have declared that no competing interests exist.

**Abbreviations:** ANC, Antenatal Care; GDM, Gestational Diabetes Mellitus; LBW, Low Birth Weight; MUAC, Mid-Upper Arm Circumference; SDG, Sustainable Development Goals; UNICEF, United Nations Children's Fund; WHO, World Health Organization.

(1.21, 6.62)] and chronically malnourished mothers [AOR = 15.3(8.12, 29.1)] were at greater risk of being stunted.

## Conclusion

Just under a third of newborns are stunted at birth, implying a pressing public health problem. Newborns born to chronically malnourished and short stature mothers were more stunted. Besides, stunting was prevalently observed among male neonates, newborns conceived in Kiremet, and being low birth weight. Thus, policymakers and nutrition programmers should work on preventing maternal undernutrition through nutrition education to reduce the burden of low birth weight and stunting. Further, paying due attention to newborns conceived in Kiremet season to improve nutritional status is recommended.

## Introduction

Stunting at birth is characterized by short length to gestational age, unlike children older than six months [1]. It's a syndrome of severe irretrievable physiological, physical, and cognitive damage due to irreversible outcome of inadequate nutrition and repeated bouts of infection that starts from conception [2–5]. As a result, children's achievement of their full economic, social, educational, and occupational potential would be compromised [6, 7]. Surprisingly, a centimeter decreases in the height of adults results in a 4% and 6% decline in wages for males and females, respectively [8]. The short and long-term shattering effects of stunting is not limited only on children's lives, but also it extends to affect the subsequent generations, too [9–13]. Stunted infants at birth are four times and two-folds to be stunted at 3 months and 2 years of age, respectively [14, 15].

Nearly, half of infant and child mortalities in Ethiopia are associated with stunting and other forms of undernutrition, which culminate in 8% decline in the country's workforce, and thus, hampering the economic growth of the nation [16]. Likewise, about 16.5% of Ethiopia's gross domestic product (GDP) is lost annually following the long-term deleterious effects of childhood stunting [17]. Further, the chronic effect of stunting ranges through acute infectious to chronic non-communicable diseases, including diarrhea, stroke, hypertension, and Diabetes Mellitus (DM).

Globally, about 144 million under-five children are suffering from stunting, of which 92% are in Low and lower-middle-income countries and Sub-Sahran countries share just over a third of the burden [18]. Even though the magnitude of stunting has shown a steady decline in all regions of the world, Africa has remained the 1st region with an escalating case [5]. Just more than three in every ten newborns develop stunting at birth, according to the INTER-GROWTH-21st development report [19].

Ethiopia is amongst other developing countries with the largest number of stunted under-five children recording an estimated 38.4% and the highest load is reported in the Amhara Region, 46% [20]. Parallelly, the 2019 Ethiopia Mini Demographic and Health Survey (EDHS) showed that stunting in children aged below six months was 17.1% [21]. Stunting at birth is attributable to multifaceted maternal and extra maternal nutritional and economic problems, including (but are not limited to) short maternal stature, poor maternal nutritional status, illness during pregnancy(DM, hypertension, and anemia), not having antenatal care (ANC) visit, born to adolescent mothers, being male, and not supplement with iron folate during pregnancy [1, 22–27].

Even though Ethiopia signed and working on the National Nutrition Policies to achieve the world health assembly's target of 26.8% stunting by 2025, the annual reduction rate of stunting is still remained at only 2.8%, which is far-off the expected reduction rate of 6% [28].

In Ethiopia, although the previous studies conducted to show stunting and its factors among under-five children, almost all studies focused on children aged between 6 and 59 months, which implie, newborn's nutritional status have been ignored, while the burden is hypothetically to be high. Various strategies are focusing on battling undernutrition among children younger than five years and these strategies have to be supported with evidence to underscore the burden as well as the contributing factors. In addition to this, continuous and updated information about early life stunting is quite imperative to reach the full vision of global policy such as reducing stunting by 40% by 2025 [29].

This study was, therefore, aimed at determining the prevalence and factors associated with stunting at birth among newborns delivered at the University of Gondar Comprehensive Specialized Referral Hospital. This will be helpful for policymakers, programmers, and researchers to reach a deliberate equity-driven policy targeting interventions for the most vulnerable population as early as possible Further, it will serve as a baseline to make comparisons with children aged 6 to 59 months to appreciate the problem among those segment of population.

## Methods

### Study design, area, and period

An institution-based cross-sectional study was conducted from February 26th to April 25th / 2020. The study was carried out at the University of Gondar Comprehensive Specialized Referral Hospital. Gondar town has one Comprehensive Specialized Referral hospital located in the North Gondar administrative zone, Amhara National Regional State, Ethiopia which is about 727 km Northwest of Addis Ababa, the capital of Ethiopia) [30]. The region has a triple burden: high prevalence of stunting; high poverty; and infrastructure development [31]. The Amhara region is the poorest of all regions, not only in Ethiopia but also in the world [32].

The University of Gondar Comprehensive Specialized Referral Hospital is a teaching Hospital that renders service for more than five million people. Moreover, the delivery ward has 3 units and the health care providers who are working in these units are 13 physicians, 50 residents, 73 midwives, and 6 General Practitioners(GP) [33].

### Study population

Alive newborns delivered at the University of Gondar Comprehensive Specialized Referral Hospital over the study period was the source population, whereas those who were delivered during the data collection period were the study population. Alive newborn-mother pairs within 72-hours of birth, during the data collection period were included. Nevertheless, extreme preterm newborns (born before 33 weeks of gestational age), were excluded. Further, newborns whose mothers suffering from critical illness (postpartum hemorrhage) and newborns suffering from birth trauma were also excluded.

### Sample size determination, sampling procedure, and technique

The sample size was determined, considering the single population proportion formula taking the following statistical assumptions: 50% proportion (p), 95% confidence intervals, 5% of margin of error, and 10% non- response rate. Finally the required sample size was 422. According to the 2019 report of the institution, about 10,599 live births were delivered at the University of Gondar Comprehensive Specialized Referral Hospital delivery ward annually.

Since the number of birth in the hospital varies throughout the year, four months were randomly drawn from the four seasons and an average birth was estimated. Then, about 884 newborns were considered as a total population(N) [34]. Finally, the study participants had been approached every other two newborn-mother paires using the systematic random sampling technique.

## Variables

**Dependent variable**. Stunting at birth (Yes/No).

**Independent variables**

**Newborn related factors**. Sex, gestational age, weight, and birth status.

**Environmental factors**. Indoor fire smoke and season of conception.

**Maternal related factors**. Height, nutritional status, illness, inter-birth interval, and iron supplementation.

## Operational definitions

**Stunting at birth**. Newborns whose length-for-gestational age below 10th percentile were deemed as stunted, whereas newborns greater than 10th percentile were defined as not stunted.

**Short maternal stature**. Mothers whose height <145cm [1].

**Maternal undernutrition**. Mothers whose MUAC measurement <22cm [35].

**Preterm**. a newborn delivered before completing 37 weeks [36].

**Term**. A newborns delivered between 37 and 42 weeks [36].

**Post-term**. A newborn delivered after completing 42 weeks [36].

**Large-for-gestational age**. Newborn's whose birth weight-for-gestational age greater than 90th percentile [36].

**Appropriate-for-gestational age (AGA)**. A newborn whose birth weight-to-gestational age fall between 10th and 90th percentile [36].

**Small weight-for-gestational age (SGA)**. A newborn whose birth weight-to-gestational age is less than 10th percentile [36].

**Short inter-birth interval**. Preceding birth interval (months between the birth of index newborn and older child)<24month (2 year) [37].

**Primigravida**. A woman got pregnant for the first time.

**Multigravida**. A woman who got pregnant for two and more times [38].

**Primiparous**. A woman who gave birth one time.

**Multipara**. A woman gave more than one birth [38].

**Anemia**. A woman whose hemoglobin measure is below 11mg/dl [39].

**Wealth status**. Using the Principal Component Analysis, participants who fall in the first, second, third, fourth, and fifth ranks were classified as richest, rich, middle, poorer, and poorest, respectively [40].

**Illness during pregnancy**. Gestational diabetes mellitus (GDM), pregnancy induce hypertension (PIH), anemia, and infection like hepatitis B, human immunodeficiency virus (HIV), and Toxoplasmosis, Rubella, Cytomegalovirus, Herpese simplex, and Syphilis (TORCHs) [23].

**Unintended pregnancy**. Mothers were asked to report whether the current pregnancy was wanted and timely. Accordingly, if mothers reported it was unwanted or mistimed, the pregnancy was deemed as 'unintended' [41].

## Data collection tools and procedures

A face-to-face structured interviewer-based administered questionnaire developed by reviewing different literature was used. In addition, chart review was implemented to gather extra

information about maternal and newborn characteristics. The newborn lie in supine recumbent position and length was measured. Two BSc midwives, one support and secure the head of the newborn and the other took measurements of the newborn's length from the top of their head to the heel of their foot. The measurement was done three times using infantometer (ITEM CODE: WS025 and SIZE18'X7'); the average length of three measurements were recorded to the nearest 0·5 cm to ensure accuracy. Similarly, the weight of newborns was measured by using a unit scale and the reading was recorded at the nearest 10g. All anthropometric measurements took place within 72 hours of birth [42, 43].

Likewise, the maternal height was measured using a wall stadiometer with a woman standing barefoot and recorded to the nearest 0·5 cm. Maternal Mid Upper Arm Circumference (MUAC) was measured by using fiber tape from the left upper arm at the mid-point between the tip of the shoulder and the tip of the elbow (olecranon process and the acromion) and the measurement was done twice to ensure its accuracy [1].

To ascertain the stunting status of the newborn, both the weight of the newborn in kilogram and the gestational age in weeks was used. The INTERGROWTH-21st standard software was used to generate a composite variable, length-for-gestational age. The length of the newborn was determined, after comparing with the same sex references. Similarly, the same procedure was followed to determine the weight of the newborn, except for considering the weight.

## Wealth status

The study participant's wealth status was assessed by using questions adopted from the 2016 EDHS report and other literature [44]. The tool comprised of the number and kinds of livestock that participants had, availability of agricultural land in hecter, the number of cereal products they gathered, the amount of money/birr available in the bank, and availability of materials in their house. The principal component analysis (PCA) was applied [40]. Then, the wealth status was ranked from the highest to the lowest.

## Data quality control

The questionnaire was pre-tested among five percent of the number of population, two weeks ahead of the actual data collection period at the University of Gondar Comprehensive Specialized Referral Hospital delivery ward. The quality of the data was further assured through careful planning and translation of the questionnaire; the English version questionnaire was translated into the local language, Amharic. To maintain the validity of the tool, its content was reviewed by senior pediatric and child health specialist nurse and nutritionist. Then, the questions were checked for its clarity, completeness, consistency, sensitiveness, and ambiguity.

A half-day training was delivered to the data collectors and supervisor aiming at briefing the objective of the study, what is/are expected from them, and so forth. Data were collected by two BSc. Midwives and supervised by one MSc nurse. Furthermore, the principal investigator and supervisor checked the collected data in daily basis for its completeness, and corrective measures were taken accordingly.

## Data processing and analysis

The collected data were entered into Epi-Info 7 version 7.2.1.0 and exported to SPSS version 20 for coding, cleaning, and analyses. Continuous independent variables were categorized.

The wealth status of mothers was analyzed through PCA and all categorical and continuous variables were categorized to be between '0' and '1'. Statistical assumptions of factor analysis such as Keiser-Meyer-Olkin (KMO) and Bartlett's test of sphericity were check. Next, all eligible factor scores were computed using the regression-based method to generate one variable,

wealth status. Following this, the final scores were ranked into five quantiles namely: first, second, third, fourth, and fifth. Finally, ranks were interpreted as richest, richer, middle, poorer, and poorest, respectively.

The outcome variable was dichotomized and coded as 0 and 1, representing those who are not stunted and stunted, respectively. For continuous variables age, for instance, the Shapiro-Wilk, statistic and histogram was used to determine the appropriate measure of central tendency. Descriptive statistics like frequency, percentage, and measures of central tendency with their corresponding measure of dispersion was used for the presentation of demographic and other independent variables. Tables and texts were used to present the findings.

Furthermore, the binary logistic regression analysis was applied to identify factors associated with stunting at birth. The Hosmer and Lemeshow test were used to diagnose the model adequacy. Variables which were failed at the chi-square and multicollinearity test were removed from multiple logistic regression analyses. The presence of Multicollinearity was examined using the Variance Inflation Factor (VIF) and a variable having larger VIF value (>5), was rejected. All variables irrespective of the significant level in the bivariables analysis were entered into the multivariable model to control the possible effects of confounder/s and to identify the significant factors. Finally, variables having independent correlations with stunting were identified and reported based on the Adjusted Odds Ratio (AOR) and p-value with its corresponding 95% CI. Further, variables having a p-value less than 0.05 were considered as statistically significant.

## Ethical considerations

Ethical clearance was obtained from the ethical review board of the school of nursing on behalf of the institutional review board of the University of Gondar. A permission letter was received from the University of Gondar Comprehensive Specialized Referral Hospital. As a S1 File, an information sheet comprised of the purpose of the study, the procedure of data collection, and the rights of the mothers was prepared and attached; it'has been reviewed and approved by the institutional review bord. After reading the information sheet to mothers, since the study didn't apply invasive procedure like body fluid sample, oral informed consent was obtained. Participants' involvement in the study were on voluntary basis and they have been told to withdraw at anytime if they wish to. All the information given by the respondents was used for the research purposes only and the confidentiality as well as privacy was maintained.

## Results

### Socio-demographic characteristics

A total of 419 newborn-mother pairs took part in the study with a 99.3% response rate. The mean (±SD) age of mothers was 27.53 (±5.5) years. A fifth 85(20.3%) of the participants were found in poorer wealth status. Regarding maternal educational status, just over a quarter 118 (28.2%) and less than a fifth 80(19.1%) of women had no schooling and had attended primary school, respectively. Close to two-thirds 274(65.4%) of mothers were housewives. Moreover, more than half of 230(54.9%) newborns were male (Table 1).

### Maternal and environmental characteristics

About 54(12.9%) of women didn't take iron supplementation during pregnancy. Just below three-fourths of (70.2%) and a third (34.1%) of women were multigravidas and primiparous, respectively (Table 2).

**Table 1. Parental and newborn socio-demographic characteristics among newborns delivered at the University of Gondar Comprehensive Specialized Referral Hospital, Northwest Ethiopia, 2020.**

| Characteristic | | Frequency | Percent (%) |
|---|---|---|---|
| **Mother's age** | 10–19 | 32 | 7.6 |
| | >35 | 60 | 14.3 |
| | 20–35 | 327 | 78.0 |
| **Residence** | Urban | 303 | 72.3 |
| | Rural | 116 | 27.7 |
| **Father's educational status** | No schooling | 102 | 24.3 |
| | Primary school | 92 | 22.0 |
| | Secondary school | 103 | 24.6 |
| | Higher education | 122 | 29.1 |
| **Mother's educational status** | No schooling | 118 | 28.2 |
| | Primary school | 80 | 19.1 |
| | Secondary school | 108 | 25.8 |
| | Higher education | 113 | 27 |
| **Mother's occupation** | Government employee | 99 | 23.6 |
| | Housewife | 274 | 65.4 |
| | Merchant | 46 | 11.0 |
| **Birth status** | Single | 407 | 97.1 |
| | Multiple | 12 | 2.9 |
| **Wealth index** | Richest | 81 | 19.3 |
| | Richer | 86 | 20.5 |
| | Middle | 83 | 19.8 |
| | Poorer | 85 | 20.3 |
| | Poorest | 84 | 20.0 |

## Anthropometric measurement

The mean birth weight of the newborns was 3059.99g (± SD 467.33g); where the median newborn's length at birth was 48 cm at 2nd percentile. The mean (±SD) maternal MUAC was 24.38cm (±2.6cm) and the median gestational age was 39 weeks (IQR = 2 weeks). The mean (±SD) height of the mothers was 154.46(±7.33). About 160 (38.2%) of mothers were chronically malnourished, whereas tiny proportion 33(7.33%) were short (Fig 1).

## Prevalence of stunting at birth

The prevalence of stunting at birth was found to be 30.5% (26.3%-35.1%).

## Factors associated with stunting at birth

In multivariable analysis, five variables have shown significant association with stunting at birth namely: sex of newborn, short stature mothers, chronically malnourished mothers, low birth weight, and season of conception.

The likelihood of being stunted among male newborns was 2.9 times higher than their female counterparts [AOR = 2.9(1.63, 5.22)]. Newborns that were, conceived in Kiremt(rainy season) and low birth weight were more likely to be stunted at birth than their counterparts by odds of 2.7 times [AOR = 2.69(1.45, 4.96)] and 3.2 times [AOR = 3.16(1.65, 6.1)], respectively.

The odds of being stunted at birth for newborns who were born to short stature mothers was 2.8 times [AOR = 2.8(1.22, 6.63)] as compared to those newborns born to tall mothers.

**Table 2. Maternal and environmental characteristics among newborns delivered at the University of Gondar Comprehensive Specialized Referral Hospital, Northwest Ethiopia, 2020.**

| Characteristic | | Frequency | Percent (%) |
|---|---|---|---|
| **Iron supplementation** | Yes | 365 | 87.1 |
| | No | 54 | 12.9 |
| **Gravidity** | Primigravida | 125 | 29.8 |
| | Multigravida | 294 | 70.2 |
| **Paritiy** | Primiparous | 143 | 34.1 |
| | Multiparous | 276 | 65.9 |
| **Short inter-birth interval** | Yes | 43 | 10.3 |
| | No | 376 | 89.7 |
| **PIH** | Yes | 37 | 8.8 |
| | No | 382 | 91.2 |
| **GDM** | Yes | 8 | 1.9 |
| | No | 411 | 98.1 |
| **Anemia** | Yes | 24 | 6.0 |
| | No | 395 | 94 |
| **ANC follow-up** | Yes | 394 | 94 |
| | No | 25 | 6.0 |
| **Gestational age** | Term | 380 | 90.7 |
| | Preterm | 39 | 9.3 |
| **Intention of pregnancy** | Yes | 387 | 92.4 |
| | No | 32 | 7.6 |
| **Season of pregnancy** | Bega (dry season) | 282 | 67.3 |
| | Kiremt(rainy season) | 137 | 32.7 |
| **Indoor fire smoke exposure** | Yes | 66 | 15.8 |
| | No | 403 | 96.2 |

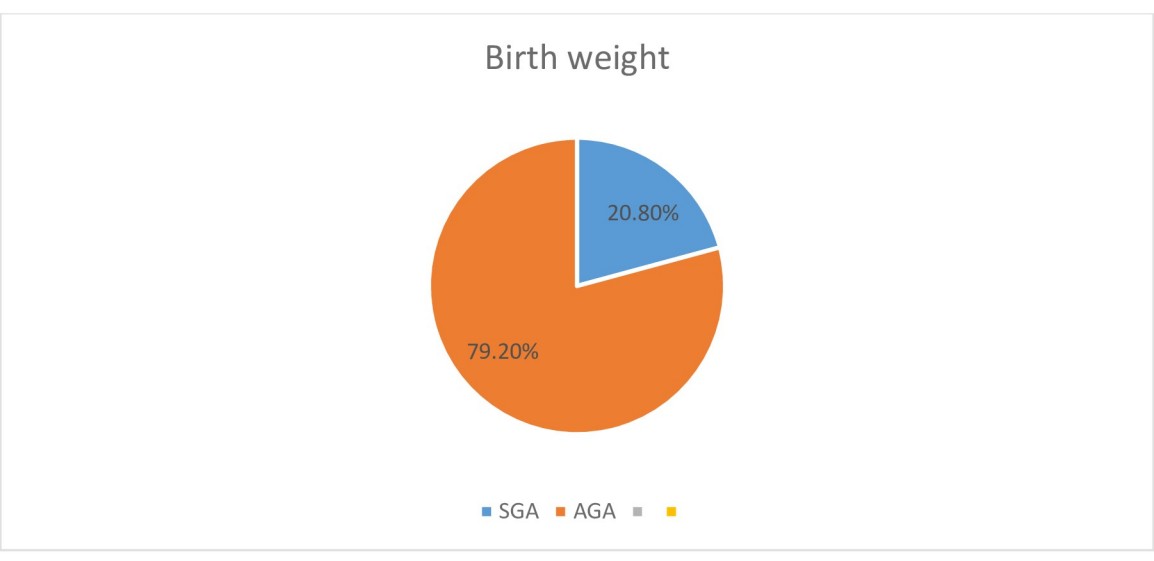

**Fig 1. Birth weight of the newborns.**

Further, chronically malnourished mothers were 15.4 times more likely to give stunted newborn than their counterparts [AOR = 15.4(8.12, 29.17)] (Table 3).

## Discussion

Stunting at birth has permanent life-threatening effects on physical, cognitive, health, and economic loss of children in their lifetime. Length at birth is the main birth outcome indicator of prenatal environment and also predictor of infant growth and survival [45].

The prevalence of stunting at birth was 30.5% (26.3%, 35.1%), depicting a very high public health problem [5], which demands the government's due emphasis. The alarming prevalence of stunting at birth reveals the seriousness of the nutritional situation in the study area. It's in line with a study done in Guatemala which is 33% [26]. It's, however, lower than a study done among seven resource-limited countries which was 43% [46]. The time gap among these studies may bring the variability, the former study was conducted in 2017. In Ethiopia, efforts has been applied over the past three years to improve the nutritional status of mothers and the maternal undernutrition is declined [47]. It's utterly known that good maternal nutritional status reduces the likelihood of neonatal stunting [48]. Besides, the current finding is higher when compared to other study conducted in Indonesia, 22.9% [49]. The economic discrepancy among the two nations could explain the observed difference although income was not associated with stunting in the current study, it's been well-established that it greatly affects the nutritional status [49]. To put simply, the Indonesian's GDP per Capita is estimated to be 4042.662 billion USD, which is higher than Ethiopian GDP Per Capita which is, 1,122.5141 USD) [50]. Likewise, Ethiopia is a third world country, especially the Amhara region where the current study is conducted, which is not only the poorest in Ethiopia but also in the world [51]. Therefore, the nation's economy is the a contributing factor for stunting at birth that the concerned body should work on.

According to the current finding male newborns were more disproportionately affected by stunting than females. The finding is congruent with a study done in Guatemala [26]. Intrinsically, programmed growth course for the male fetus is greater than that of females, which results in higher demands for most nutrients, i.e. males respond with minimal gene and protein changes in the placenta with continued growth in a suboptimal intrauterine environment, which puts them at higher risk [52]. On the other hand, females express multiple placental genes and protein changes that result in a milder decrease in growth without actual growth restriction [53]. Furthermore, other than the placental growth, male fetuses are liable for adverse events than female associated with the rapid body and brain growth [54].

The likelihood of being stunted among newborns who were conceived in Kiremt(rainy) season was 2.7 times higher than newborns conceived in Belg season, which is consistent with a study employed in Bangladeshi [55], this suggested that numerous mothers who conceived in Kiremt season are more prone to giving birth to stunted newborns. This could be: i)in Ethiopia this season is the toughest time when food insecurity is commonly observed, especially among rural residents [56]; ii) Infectious disease outbreak like a diarrheal disease is highly prevalent [56]; and iii)since 85% of the Ethiopian population is farmer there is the highest workload in this season [57, 58]. Food insecurity linked to low intake of nutritious diet during pregnancy [59]. Similarly, high workload during pregnancy and working long hours increases extra metabolic expenditure, further reducing the fetal size, as sweating reduces plasma volume(decrease uteroplacental blood flow) and circulatory blood flow in the uterus [60, 61]. This suggests that stakeholders shall better work on infection prevention and nutrition interventions in Kiremet(rainy) season. In addition, mothers need to be counseled to minimize workload during pregnancy.

**Table 3. Bivariable and multivariable logistic regression output, showing that factors associated with stunting at birth among newborns in Gondar Comprehensive Specialized Referral Hospital, Northwest Ethiopia.**

| Characteristic | Stunting at birth | | COR(95%CI) | AOR(95%CI) | P-value |
|---|---|---|---|---|---|
| | Stunted | Not stunted | | | |
| **Age of mother** | | | | | |
| 15–19 | 15(3.57%) | 17(4.06%) | 2.09(1.004,4.358) | 1.390(0.468, 4.127) | 0.55 |
| >35 | 16(3.81%) | 44(10.50%) | 0.862(0.464,1.602) | 0.653(0.272, 1.568) | 0.34 |
| 20–35 | 97(23.15%) | 230(54.89%) | 1 | 1 | |
| **Residence** | | | | | |
| Rural | 45(10.74) | 71(16.94) | 1.680(1.070,2.637) | 0.995(0.430, 2.301) | 0.99 |
| Urban | 83(19.81) | 220(52.50) | 1 | 1 | |
| **Religion** | | | | | |
| Muslim | 14(3.34) | 45(10.74) | 0.671(0.354,1.273) | 0.691(0.294,1.621) | 0.39 |
| Orthodox | 114(27.21) | 246(58.71) | 1 | 1 | |
| **Father's educational status** | | | | | |
| No schooling | 36(8.59%) | 66(15.75%) | 1.831(1.020,3.289) | 1.321(0.394,4.429) | 0.65 |
| Primary | 29(6.9%) | 63(15.03%) | 1.545(0.840,2.843) | 0.857(0.305,2.404) | 0.76 |
| Secondary | 35(8.35%) | 68(16.23%) | 1.728(0.961,3107) | 1.109(0.457,2.689) | 0.81 |
| Higher education | 28(6.68%) | 94(22.43%) | 1 | 1 | |
| **Mother's educational status** | | | | | |
| No schooling | 42(10.02%) | 76(18.13%) | 1.849(1.038,3.296) | 0.48(0.118,2.018) | 0.32 |
| Primary school | 23(5.48%) | 57(13.60%) | 1.350(0.703,2.594) | 0.600(0.171, 2.107) | 0.42 |
| Secondaryschool | 37(8.83%) | 71(16.94%) | 1.744(0.965,3.150) | 0.868(0.309, 2.440) | 0.78 |
| Higher education | 26(6.20%) | 87(20.76%) | 1 | 1 | |
| **Mother's occupation** | | | | | |
| Government employee | 20(4.77%) | 79(18.85%) | 1 | 1 | |
| Housewife | 99(23.62%) | 175(41.76%) | 2.235(1.290,3.869) | 2.312(0.796,6.711) | 0.12 |
| Merchant | 9(2.15%) | 37(8.83%) | 0.961(0.399,2.312) | 0.996(0.275,3.607) | 0.99 |
| **Wealth index** | | | | | |
| Richest | 28(6.68%) | 53(12.64%) | 1 | 1 | |
| Richer | 32(7.63%) | 54(12.88%) | 1.122(0.596,2.113) | 1.142(0.494,2.639) | |
| Middle | 28(6.68%) | 55(13.13%) | 0.964(0.505,1.838) | 1.470(0.622,3.476) | |
| Poorer | 27(6.44%) | 58(13.84%) | 0.881(0.421,1.412) | 0.787(0.316,1.963) | |
| Poorest | 13(3.10%) | 71(16.94%) | 0.558(0.301,1.033) | 0.488(0.181,1.318) | |
| **Sex of newborn** | | | | | |
| Male | 81(19.33%) | 149(35.56%) | 1.642(1.072,2.516) | 2.916(1.629,5.218) | 0.00 |
| Female | 47(11.21%) | 142(33.89%) | 1 | 1 | |
| **Take iron during pregnancy** | | | | | |
| **Yes** | 108(25.77%) | 257(61.34%) | 1 | 1 | |
| **No** | 20(4.77%) | 34(8.11%) | 1.400(0.771,2.541) | 1.398(0.494,3960) | 0.52 |
| **Parity** | | | | | |
| Multipara | 81(19.33%) | 195(46.54%) | 0.848(0.549,1.31) | 1.332(0.690,1.571) | 0.39 |
| Primiparous | 47(11.21%) | 96(22.91%) | 1 | 1 | |
| **ANC follow-up** | | | | | |
| Yes | 119(28.40%) | 275(65.63) | 1 | 1 | |
| No | 9(2.14%) | 16(3.82%) | 1.300(1.44,8.00) | 0.428(0.108,1.688) | 0.22 |
| **Intention of pregnancy** | | | | | |
| Not intentional | 13(3.10%) | 19(4.53%) | 1.618(0.773,3.386) | 1.109(0.383,3.213) | |
| Intentional | 115(27.45%) | 272(64.91%) | 1 | 1 | 0.84 |

*(Continued)*

**Table 3.** (Continued)

| Characteristic | Stunting at birth | | COR(95%CI) | AOR(95%CI) | P-value |
|---|---|---|---|---|---|
| | Stunted | Not stunted | | | |
| **Mother's height** | | | | | |
| Short | 28(6.68%) | 17(4.06%) | 4.513(2.369,8.599) | 2.841(1.218, 6.626) | 0.16 |
| Tall | 100(23.86%) | 274(65.39%) | 1 | 1 | |
| **Nutritional status** | | | | | |
| Malnourished | 95(22.67%) | 65(15.51%) | 0.01(6.177,16.22) | 15.39(8.12,29.17) | 0.00 |
| Normal | 33(7.87%) | 226(53.93%) | 1 | 1 | |
| **Birth weight** | | | | | |
| SGA | 43(10.26%) | 44(10.50%) | 2.840(1.745,4.623) | 3.158(1.645, 6.061) | 0.00 |
| AGA | 85(20.28%) | 247(58.94%) | 1 | 1 | |
| **Season of conception** | | | | | |
| Kiremt (rainy) | 45(10.74%) | 92(21.96%) | 1.173(0.756,1.819) | 2.691(1.455,4.977) | 0.00 |
| Belg (dry) | 83(19.81%) | 199(47.49%) | 1 | 1 | |
| **Exposed to indoor fire smoke** | | | | | |
| Yes | 24(5.73%) | 42(10.02%) | 1.368(0.788,2.374) | 0.866(0.414, 1.816) | 0.70 |
| No | 104(24.82%) | 249(59.43%) | 1 | 1 | |

1 = Reference category.

The odds of being stunted among low birth weight newborns was 3.1 time higher as compared to normal weight, which is in agreement with a study done in Low- and middle-income countries [62]. The reason could be: apart from genetic reason, LBW is the signal of premature birth or Intra Uterine Growth Restriction [63].

The odds of being stunted at birth among newborns born from chronically malnourished mothers was higher than their counterparts. This finding is in line with a study reported from Indonesia [64, 65]. Maternal nutrition affects fetal growth directly by determining the number of nutrients available indirectly by affecting the fetal endocrine system, and epigenetically by modulating gene activity [66]. Normal pregnancy entails substantial production of hormones in the maternal, placental, and fetal compartments. The secretion of most important hormones like glucocorticoids, insulin-like growth factors, and leptin, can be affected by maternal undernutrition that could further affect fetal development and growth [67]. This event informed that lots of mothers who are chronic malnutritioned imposed to give stunted birth. In this way, undernutrition passes from one generation to another as a grim inheritance [68]. Maternal nutrition can be considered as one of the major contributing factors for stunting at birth and that the responsible bodies should invest on food intervention during pregnancy to prevent it's short & long-term effect/s of stunting. However, a study conducted in East Java, Indonesia maternal chronic energy deficiency was not associated with stunting at birth [1]. The difference could be: Indonesia's health expenditure is higher than the Ethiopia [69].

Short stature mothers were more likely to give stunted newborns than their counterparts. The finding is in congruent with those studies piloted in different countries [1, 26, 64, 70]. Since short stature is an indicator of chronic malnutrition and environmental insult exposure from fetal life to adulthood, illustrated that short stature mothers lack a well-developed uterus given that poor nutrient storage [64] and narrow pelvic(cephalo-pelvic disproportion), this leads to fetal growth continue with limited nutrition and the narrowed pelvic results in faltering fetal growth [62], as the growth of the baby starts from the period mid-gestation [71]. This finding suggests combating stunting during the childhood period enables to create stunting

free generations. Moreover, stunting in the earlier critical period would result in stunting in the later period on the one hand and over nutrition on the other hand [72]. This indicates that stunting during childhood period results in double burden of malnutrition that the world, especially developing countries are facing [5]. This strongly shows that stunting in newborns and mothers had intergenerational nature of the problem (Inherited from mothers to their babies), which needs urgent intervention on the provision of female nutrition, prevention of infectious disease like diarrheal.

The study uses a composite variable (length-for-gestational age) to determine stunting at birth unlike previous studies used just only the length measurement regardless of the gestational age. On the other hand, this study didn't assess pre-pregnancy weight, which is the major contributing factor of maternal nutrition and the study shares the limitation of the cross-sectional study. Further, though various strategies that have been applied to control bias like recall biases, the study still shares admissible shortcomings of bias.

## Conclusion

Stunting at birth is a very high public health problem, demanding a grave measure. Newborns born to chronically malnourished and short stature mothers are more stunted. Likewise, stunting is prevalently observed among newborns conceived in Kiremet (rainy season), being male, and, being low birth weight. Thus, dealing with maternal malnutrition through provision of nutrition education and promoting nutrient intake is recommended. Policymakers and nutrition programmers should work on identified attributes like reducing the magnitude of short maternal stature through tacking childhood and adolescence undernutrition so as to decrease the magnitude of stunting a birth. Due stress shall be given for those mothers who conceive in Kiremt (rainy) season through prevention of diarrheal and other infectious diseases. Longitudinal studies are recommended to assess whether stunting at birth will go through their lifetime, or not. Further, cohort study is recommended to show the true association between maternal undernutrition and stunting at birth.

## Supporting information

**S1 File.**
(DOC)

## Acknowledgments

We would like to thank the data collectors and faculties who have contributed to this work.

## Author Contributions

**Conceptualization:** Almaz Tefera Gonete, Eskedar Getie Mekonnen, Wubet Worku Takele.

**Data curation:** Almaz Tefera Gonete, Bogale Kassahun, Wubet Worku Takele.

**Formal analysis:** Almaz Tefera Gonete, Eskedar Getie Mekonnen, Wubet Worku Takele.

**Funding acquisition:** Bogale Kassahun.

**Investigation:** Bogale Kassahun.

**Methodology:** Almaz Tefera Gonete, Eskedar Getie Mekonnen, Wubet Worku Takele.

**Project administration:** Bogale Kassahun.

**Resources:** Bogale Kassahun.

**Software:** Almaz Tefera Gonete, Wubet Worku Takele.

**Supervision:** Almaz Tefera Gonete, Bogale Kassahun, Eskedar Getie Mekonnen.

**Validation:** Wubet Worku Takele.

**Visualization:** Wubet Worku Takele.

**Writing – original draft:** Almaz Tefera Gonete, Bogale Kassahun, Eskedar Getie Mekonnen, Wubet Worku Takele.

**Writing – review & editing:** Eskedar Getie Mekonnen, Wubet Worku Takele.

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
