## [Decision Letter · Decision Letter 0]

17 Dec 2020

PONE-D-20-32536

Stunting at birth and associated factors among newborns delivered at the University of Gondar Comprehensive Specialized Referral Hospital.

PLOS ONE

Dear Dr. Gonete,

Thank you for submitting your manuscript to PLOS ONE. After careful consideration, we feel that it has merit but does not fully meet PLOS ONE’s publication criteria as it currently stands. Therefore, we invite you to submit a revised version of the manuscript that addresses the points raised during the review process.

This is an interesting and largely well-written study. The reviewers have suggested a number of revisions that need to be made, all of which are important, which I will not add to. I look forward to receiving a revised version of the manuscript.

We look forward to receiving your revised manuscript.

Kind regards,

Clive J Petry, PhD

Academic Editor

PLOS ONE

Journal Requirements:

2. Please provide citations for the literature used to develop the survey, and describe how the newly constructed survey was validated.

3. Please include additional information regarding the survey or questionnaire used in the study and ensure that you have provided sufficient details that others could replicate the analyses. For instance, if you developed a questionnaire as part of this study and it is not under a copyright more restrictive than CC-BY, please include a copy, in both the original language and English, as Supporting Information.  If the original language is written in non-Latin characters, for example Amharic, Chinese, or Korean, please use a file format that ensures these characters are visible.

4. Please amend your current ethics statement to address the following concerns: Please explain why written consent was not obtained, how you recorded/documented participant consent, and if the ethics committees/IRBs approved this consent procedure.

"The University of Gondar sponsored the study, however, the funder didn’t have a role in the study"

"yes, but the funder has no role in the study design, data collection and analysis, decision to publish, or preparation of the manuscript"

7. Please include a copy of Table 3 which you refer to in your text on page 15.

8. We note you have included a table to which you do not refer in the text of your manuscript. Please ensure that you refer to Table 4 in your text; if accepted, production will need this reference to link the reader to the Table.

Reviewers' comments:

Reviewer's Responses to Questions

**Comments to the Author**

1. Is the manuscript technically sound, and do the data support the conclusions?

Reviewer #1: Yes

Reviewer #2: Yes

2. Has the statistical analysis been performed appropriately and rigorously? 

Reviewer #1: Yes

Reviewer #2: Yes

3. Have the authors made all data underlying the findings in their manuscript fully available?

Reviewer #1: Yes

Reviewer #2: No

4. Is the manuscript presented in an intelligible fashion and written in standard English?

Reviewer #1: Yes

Reviewer #2: No

5. Review Comments to the Author

Reviewer #1: Comments_1: Authors wrote unnecessarily a lot about background profile of the study, please make it short, simple and precise.

Comments_2: Are you sure no studies were available that highlighted stunted children at age 0 months? As all DHS reports this group of children.

Comments_3: Need a major revision in the Literature review section.

Comments_4: Write ‘This’ instead of ‘The’ in line number 100.

Comments_5: Specify the name of the region in line number 119.

Comments_6: Methods section is too long to read. Authors could make it short by combining Data Collection tools and procedures, Data quality control and Data processing and analysis and removing redundant information.

Comments_7: According to WHO when a baby is born before 28 weeks of gestational age then it is known as extremely preterm. Why did you exclude the extreme preterm newborns (born before 33 weeks of gestational age)?

Comments_8: Give the reason(s) of excluding the newborns whose mothers suffering from critical illness (postpartum hemorrhage) and newborns suffering from an illness (birth trauma)? Would these factors be not the reason for the stunting at birth?

Comments_9: Did the authors or institution collect the data?

Comments_10: Was the data primary or secondary? Make it clear.

Comments_11: What was your actual study period? It’s confusing, as your data collection period and study period is different.

Comments_12: What was the actual sample size? In method section it isn’t clearly stated. In abstract it is 422, but in result section it is 419. So this is puzzling.

Comments_13: Mention the year beside the months in line 141.

Comments_14: In line 196, the authors described about wealth index. But in the Data processing and analysis section they again described about income status of mothers is a bit confusing.

Comments_15: The lines 242 and 243 are confusing, why fisher exact test was performed.

Comments_16: Give an acceptance/rejection range for Variance Inflation Factor (VIF) in line 246.

Comments_17: VIF is used to assess the multicollinearity for multiple linear regression analysis. How it works for multiple logistic regressions analyses please explain.

Comments_18: You should display the results of all the study variables using tables, graphs, charts, or figures.

Comments_19: In table 1, specify the age range (beside the categories) of the mothers.

Comments_20: Use the table/figures to represent the Maternal Characteristics and Environmental characteristics of the newborns.

Comments_21: Change the table number in line 383.

Comments_22: Add an extra column in table 3 to indicate the p-values.

Comments_23: Add the meaning of ‘1’ in notes section under the table 3.

Comments_24: In line 405, you said that the overall stunting at birth was 41.7 % (37% - 46.7%). What’s the basis of this statement? Is this your findings?

Comments_25: As income is not associated with the stunting then why authors made the recommendation.

Comments_26: It is pretty obvious that malnourished mothers will give low birth weight children in most cases that causes stunting/malnutrition among new born and the findings showed high correlation why authors tried to justify this findings?

Comments_27: Policy recommendation is missing in this study.

Comments_28: Provide a list of abbreviation.

Comments_29: Add the reference(s) in line number 71, 89, 91, 97, 124, 126, 140, 197, 226, 227, 249, and 420.

Comments_30: Correctly cite the reference in line number 127.

Reviewer #2: Explanation for the Q number:

3. The data about low birth weight was not presented. only small for gestational age (SGA), that's not the same with low birth weight. There are some variables that are not described in the operational definition/objective criteria:

- classification of the mother's age

- Intention of pregnancy

- malnourished mother (what criteria that was used for determining malnourished; MUAC?, it should be described.

- please consider the definition of stunting at birth, because the length of birth is associated with gestational age. The length <48 cm maybe only for a term newborn in general, while for preterm we have to look at the growth chart that we used. if the length for gestational age is appropriate, we cannot say that is stunted newborn.

Preferably data on the mean or median birth length of the stunted newborn are presented.

4. The are several typing errors, and abbreviations that are not common (Px; pregnancy)

Tables: it is not common to put the size of the samples (n) on the title of the table

Discussions: Semarang is one of the city in Indonesia, so it cannot be written as two different cities or country. please check the citation.

6. PLOS authors have the option to publish the peer review history of their article (what does this mean?). If published, this will include your full peer review and any attached files.

Reviewer #1: **Yes: **Md. Shariful Islam, Lecturer, Dept. of Public Health, First Capital University of Bangladesh, Chuadanga, Khulna, Bangladesh

Reviewer #2: No

---

## [Author Response · Author response to Decision Letter 0]

25 Dec 2020

December 24/2020

Point-by-point response round one 

Dear both the editor and reviewers, we found your comments are to be crucial for enhancing the tone and readability of our scholarly work. We are really grateful enough to express our sincerest thanks for your comment. Appreciating your effort and valuable comments, we have provided possible reflections on the raised concerns and questions. Kindly find our reflections hereunder. 

A. Editor’s comment 

1. Please provide citations for the literature used to develop the survey, and describe how the newly constructed survey was validated.

Authors’ response: Comment has been accepted.

2. Please include additional information regarding the survey or questionnaire used in the study and ensure that you have provided sufficient details that others could replicate the analyses. For instance, if you developed a questionnaire as part of this study and it is not under a copyright more restrictive than CC-BY, please include a copy, in both the original language and English, as Supporting Information. If the original language is written in non-Latin characters, for example Amharic, Chinese, or Korean, please use a file format that ensures these characters are visible.

Authors’ response: Dear, the outcome variable is ascertained using an objective composite variable (newborn length and gestational age). For the independent variables, we had collected from various published articles, and variables that were not included previously and assume to have an association were also included. We have supplied a copy of both Amharic (local language) and English language tool that we used to collect the data. 

3. Please amend your current ethics statement to address the following concerns: Please explain why written consent was not obtained, how you recorded/documented participant consent, and if the ethics committees/IRBs approved this consent procedure.

Authors’ response: Comment accepted and amendment has been done.

4. Thank you for stating the following in the Acknowledgments Section of your manuscript:"The University of Gondar sponsored the study, however, the funder didn’t have a role in the study" We note that you have provided funding information that is not currently declared in your Funding Statement. However, funding information should not appear in the Acknowledgments section or other areas of your manuscript. We will only publish funding information present in the Funding Statement section of the online submission form. Please remove any funding-related text from the manuscript and let us know how you would like to update your Funding Statement. Currently, your Funding Statement reads as follows:

Authors’ response: Dear, we have checked the acknowledgment section, but we didn’t find a statement that describes funding information. We have deleted the funding statement in the manuscript and we are grateful to use your modification on our behalf.

B. Reviewer #1

1. Authors wrote unnecessarily a lot about background profile of the study, please make it short, simple and precise.

Authors’ response: We found the comment is to be relevant and modification has been done accordingly.

2. Are you sure no studies were available that highlighted stunted children at age 0 months? As all DHS reports this group of children.

Authors’ response: As apparently seen in our background section, the prevalence of children younger than six months is 17.1%, but they determined using the World Health Organization’s classification (z-score) considering their age and height, which is different from our method of ascertainment and not applicable for children at age 0 months. We are not quite sure they included children at birth. Moreover, factors were not explored in that report. Therefore, considering the abovementioned issues, we opted to declare no study in that age group.

3. Need a major revision in the Literature review section.

Authors’ response: Comment found to be valid and correction has been done accordingly.

4. Write ‘This’ instead of ‘The’ in line number 100.

Authors’ response: comment accepted!

5. Specify the name of the region in line number 119.

Authors’ response: comment accepted and corrected accordingly. 

6. Methods section is too long to read. Authors could make it short by combining Data Collection tools and procedures, Data quality control and Data processing and analysis and removing redundant information.

Authors’ response: comment accepted and change has been done. 

7. According to WHO when a baby is born before 28 weeks of gestational age then it is known as extremely preterm. Why did you exclude the extreme preterm newborns (born before 33 weeks of gestational age)?

Authors’ response: We used software to analyze the outcome variable (stunting) and birthweight prepared by the INTERGROWTH-21st project. The standard instructs users to not consider newborns aged less than 33 weeks. We, therefore, excluded that segment of the population in order not to introduce bias.

8. Give the reason(s) of excluding the newborns whose mothers suffering from critical illness (postpartum hemorrhage) and newborns suffering from an illness (birth trauma)? Would these factors be not the reason for the stunting at birth?

Authors’ response: We had excluded both mothers and newborns having health problems during data collection in order not to breach ethical issues. From a research ethics point of view, it’s highly recommended that the study participants should be as stable as possible and healthy to capture the necessary data. In addition to this, we may increase the non-response rate and introduce bias and eventually reach false inference. During the data collection period, we had encountered women having a postpartum hemorrhage (PPH) and a baby sustained birth trauma following instrumental delivery. Measuring the newborn’s body dimension while the baby is managed in a neonatal intensive care unit (NICU) would lead to further medical problems like sepsis and hypothermia beyond the ethical concern. Dear, we don’t see the connection between the aforementioned factors and the outcome variable. Indeed, we don’t believe that neither the burden of stunting among these populations is higher nor these factors expose to stunting.

9. Did the authors or institution collect the data?

Authors’ response: Not really! As can be seen in the data collection section, the data was gathered and supervised by well-trained midwives, who are not investigators of the study. Nevertheless, even though authors didn’t involve directly in the data collection task, we had been supervising the overall activity so as to maintain the quality of the data. Further, we had delivered the training to both data collectors and supervisors.

10. Was the data primary or secondary? Make it clear.

Authors’ response: In the data collection tool and procedures section, it read as “both interview and chart review was done”, implying we used a mixed data collection method (primary and secondary).

11. What was your actual study period? It’s confusing, as your data collection period and study period is different.

Authors’ response: Our data collection period was, from February 26th through April 25th / 2020. Dear, we have reviewed the document, and we did find nothing a different data collection period other than the above-stated period. Probably, the statement that appeared under the sample size determination, procedure, and technique section would be confusing. ‘Four months were randomly drawn from the four seasons; accordingly, around 839, 924, 954, and 816 live births were delivered in September, December, May, and August yielding an average birth (N) of 884 newborns’. It’s not to refer to the study period or data collection period. We had collected the number of births over the aforementioned months to estimate the general population, as the number varies across seasons in the Ethiopian context. In a nutshell, we just use the average number of neonates delivered in those months to determine the total population in order to estimate the ‘K’ interval. The data collection/study period is from February 26th through April 25th / 2020. 

12. What was the actual sample size? In method section it isn’t clearly stated. In abstract it is 422, but in result section it is 419. So this is puzzling.

Authors’ response: It’s really a wonderful concern! In the sample size and determination section, we have provided the detail of statistical assumptions considered to estimate the required sample size, however, we did miss to put the final sample size. The actual sample size was 422, of which 419 participants responded and included in the final analysis, making a response rate of 99.3%. Dear, now we’ve added the exact sample size in the sample size determination section. All in all, we don’t recognize any fallacy regarding the sample size.

13. Mention the year beside the months in line 141

Authors’ response: Comment accepted and corrected accordingly.

14. In line 196, the authors described about wealth index. But in the Data processing and analysis section they again described about income status of mothers is a bit confusing.

Authors’ response: Dear, yes! We used principal component analysis (PCA) to determine the wealth status of the participants. Comment accepted, and we’ve replaced the ‘income status’ phrase with ‘wealth status’.

15. The lines 242 and 243 are confusing, why fisher exact test was performed.

Authors’ response: Definitely, it’s confusing! We’ve checked the chi-square and multicollinearity assumptions before running the multivariable analysis; we didn’t consider fisher exact test as it’s not a required statistical assumption. We have removed that in the newly revised document.

16. Give an acceptance/rejection range for Variance Inflation Factor (VIF) in line 246.

Authors’ response: comment has been accepted and a statement is added. 

17. VIF is used to assess the multicollinearity for multiple linear regression analysis. How it works for multiple logistic regressions analyses please explain.

Authors’ response: Indeed, VIF is used to estimate the correlation of variables and it’s usually applicable for linear regression and it’s one of the assumptions required to be fulfilled prior to run the analysis. However, it’s also applicable in multivariable logistic regression; here as you know the aim is distinct from linear regression. One of the statistical assumptions to be fulfilled before running logistic regression is that, there should not be linearity/correlation between the independent variables that are going to be considered to see its association with the outcome variable and VIF is the diagnostic method. Generally, here, our aim is to see the correlation within the independent variables, not with the outcome variable, unlike linear regression.

18. You should display the results of all the study variables using tables, graphs, charts, or figures.

Authors’ response: Comment accepted, and we’ve used a table and figure to present our data, especially for those variables, which were presented using text only. 

19. In table 1, specify the age range (beside the categories) of the mothers.

Authors’ response: We’ve modified, and presented the age range only. Dear, here, our concern is to examine the association of extreme maternal age (adolescent (10-19) and advanced age (>35)) with stunting, considering the truth that extreme age is a risk factor to give birth to stunted newborn as compared to mothers aged between 20 and 35 years. Initially, we were labeled the maternal age as ‘adolescent’ ‘young’, and ‘old/advanced’. There is no problem with operationalize ‘adolescent’ and ‘old/advanced’ age as there are established and standard definitions for both, however, there is no definition for mothers age from 20-35 years. Other previous scholars also put the age range, not the interpretation (http://theicph.com/wp-content/uploads/2018/04/11.-VIVIN-EKA-RAHMAWATI.pdf). 

Therefore, to be consistent and to avoid the confusion, we just delete the interpretation and put the age range, instead.

20. Use the table/figures to represent the Maternal Characteristics and Environmental characteristics of the newborns.

Authors’ response: Comment accepted and amendment has been done!

21. Change the table number in line 383.

Authors’ response: Comment accepted!

22. Add an extra column in table 3 to indicate the p-values.

Authors’ response: Dear, we did intentionally ignore to add one column to present the p-values estimate and use the asterisk (*) to show the degree of association in order not to make the table bulk, which we believe it’s scientifically acceptable. Anyways, we have added a column and put the respective as per your comment.

23. Add the meaning of ‘1’ in notes section under the table 3.

Authors’ response: Comment accepted!

24. In line 405, you said that the overall stunting at birth was 41.7 % (37% - 46.7%). What’s the basis of this statement? Is this your findings?

Authors’ response: Dear, your comment is appreciated and accepted. We have noticed the mistake after submitting the manuscript to the journal. We had uploaded the old version of our manuscript. We’ve made changes to the manuscript, especially in the second paragraph of the discussion section. The true finding is the one that appeared both in the abstract and results section i.e. 30.5 %( 95% CI: 26.3%, 34.8%), not 41.7 % (37% - 46.7%). In a nutshell, we’ve revised and made change to the second paragraph of the discussion section. 

25. As income is not associated with the stunting then why authors made the recommendation.

Authors’ response: You are right that income doesn’t associate with stunting in our study, and in the discussion section we have declared that the variable is not found to be significant, nevertheless, that doesn’t mean it has no effect on nutritional status. Dear, we believe that relying only on statistical association and give inference might not be always good, as statistics is not always certain. We’ve to think of this gap and look at the scientific connection that income has on nutritional status. Further, it’s recommended to discuss important variables even though that doesn’t show an association just to give emphasis to the issue in order to enforce policymakers not to ignore the issue assuming the finding is disputable across various studies in the field.

26. It is pretty obvious that malnourished mothers will give low birth weight children in most cases that causes stunting/malnutrition among newborn and the findings showed high correlation why authors tried to justify this findings?

Authors’ response: You are quite right that the effect of maternal malnutrition on child stunting is a well-established truth and other studies witnessed similar finding in this regard. However, keeping the common justification given to the association by different scholars in the field, there are also important justification points that have been missed by some scholars, making justification variable across different studies in the field. We believe that showing the possible justifications to the observed association would strengthen the other’s justification given by different scholars, and urge the policymakers to counteract the problem.

27. Policy recommendation is missing in this study.

Authors’ response: Dear, we see that your comment is critical and in the conclusion section we’ve forwarded recommendations to policymakers.

28. Provide a list of abbreviation.

Authors’ response: Comment accepted and provided!

29. Add the reference(s) in line number 71, 89, 91, 97, 124, 126, 140, 197, 226, 227, 249, and 420.

Authors’ response: Comment accepted and cited!

30. Correctly cite the reference in line number 127.

Authors’ response: Comment accepted!

C. Reviewer #2

1. The data about low birth weight was not presented. Only small for gestational age (SGA), that's not the same with low birth weight.

Authors’ response: You are right that the prevalence of low birth weight was not presented, rather only the mean birth weight was recorded. Dear, as we’ve included preterm births in our study, we didn’t use the criteria used for term newborns i.e <2500g to define low birth weight, we instead applied birthweight adjusted to gestational age regardless of the maturity level of the newborn, and classified as LGA, SGA, and AGA using the INTERGROWTH-21st reference guideline. Finally, we reported the SGA’s estimate as LBW.

2. Classification of the mother's age.

Authors’ response: We’ve modified and presented the age range only. Dear, here, our concern is to examine the association of extreme maternal age (adolescent (10-19) and advanced age (>35)) with stunting, considering the truth that extreme age groups (old/advanced and/or adolescent) are/is a risk factor to give birth to stunted newborns as compared to mothers aged between 20 and 35 years. Initially, we had labeled the maternal age as ‘adolescent’ ‘young’, and ‘old/advanced’. There is no problem to operationalize ‘adolescent’ and ‘old/advanced’ age as there are established and standard definitions for both, however, there is no definition for mothers’ age from 20-35 years, who are considered to be safe groups to give birth to stunted babies. Therefore, to avoid confusion, we just deleted the interpretation and put the age range instead. 

3. Define intention of pregnancy

Authors’ response: Comment accepted and the variable has been defined and added in the operational definition section.

4. Malnourished mother (what criteria that was used for determining malnourished; MUAC?, it should be described

Authors’ response: Dear, we do define maternal malnutrition based on MUAC measurement. As can be seen in the optional definition and data collection procedures section, the mother’s MUAC measurement <23.5 was defined as ‘malnourished’. Actually, in that section, the variable is read as ‘chronic energy deficiency’, which is the usual and appropriate interpretation for MUAC measurements. We thus keep the variable name as it’s, however, we are ready to change the naming in the next revision if it’s your recommendation.

5. Please consider the definition of stunting at birth, because the length of birth is associated with gestational age. The length <48 cm maybe only for a term newborn in general, while for preterm we have to look at the growth chart that we used. If the length for gestational age is appropriate, we cannot say that is stunted newborn.

Preferably data on the mean or median birth length of the stunted newborn are presented.

Authors’ response: Your concern and reflection is quite admissible. Similar to the birth weight of the newborn, we defined stunting considering the gestational age and analyzed using the INTERGROWTH-21st software prepared to determine the birthweight and length of the newborn considering the gestational age. As can be seen in the operational definition section, we defined using newborns length and gestational age to generate percentile, not simply by measuring only the length of the newborn. 

Taking all together, we don’t define using the simple measurement of length and labeled as ‘stunted’ and ‘not stunted’, we instead use a growth reference chart that considers gestational age and length irrespective of the maturity level of the newborns(term and preterm), according to the INTERGROWTH-21st project’s recommendation. We didn’t use (<48cm) to define stunting for term newborns. 

6. The are several typing errors, and abbreviations that are not common (Px; pregnancy)

Authors’ response: Comment accepted!

7. Tables: it is not common to put the size of the samples (n) on the title of the table

Authors’ response: Comment accepted and revision has been done accordingly.

8. Discussions: Semarang is one of the city in Indonesia, so it cannot be written as two different cities or country. Please check the citation.

Authors’ response: Comment accepted and corrected accordingly.

 Thank you very much for your time!

---

## [Editor Report · Decision Letter 1]

4 Jan 2021

Stunting at birth and associated factors among newborns delivered at the University of Gondar Comprehensive Specialized Referral Hospital.

PONE-D-20-32536R1

Dear Dr. Gonete,

We’re pleased to inform you that your manuscript has been judged scientifically suitable for publication and will be formally accepted for publication once it meets all outstanding technical requirements.

Kind regards,

Clive J Petry, PhD

Academic Editor

PLOS ONE
---

## [Editor Report · Acceptance letter]

8 Jan 2021

PONE-D-20-32536R1 

Stunting at birth and associated factors among newborns delivered at the University of Gondar Comprehensive Specialized Referral Hospital. 

Dear Dr. Gonete:

I'm pleased to inform you that your manuscript has been deemed suitable for publication in PLOS ONE. Congratulations! Your manuscript is now with our production department. 

Kind regards, 

on behalf of

Dr. Clive J Petry 

Academic Editor

PLOS ONE